# Energetic Aspects of Turfgrass Mowing: Comparison of Different Rotary Mowing Systems

**Michel Pirchio \*, Marco Fontanelli, Fabio Labanca, Mino Sportelli, Christian Frasconi, Luisa Martelloni[ID], Michele Raffaelli, Andrea Peruzzi, Monica Gaetani, Simone Magni, Lisa Caturegli[ID], Marco Volterrani and Nicola Grossi[ID]**

Department of Agriculture, Food and Environment, University of Pisa, 56124 Pisa, Italy
\* Correspondence: michel.pirchio@for.unipi.it

**Abstract:** Turfgrass mowing is one of the most important operations concerning turfgrass maintenance. Over time, different mowing machines have been developed, such as reel mowers, rotary mowers, and flail mowers. Rotary mowers have become the most widespread mowers for their great versatility and easy maintenance. Modern rotary mowers can be equipped with battery-powered electric motors and precise settings, such as blade rpm. The aim of this trial was to evaluate the differences in power consumption of a gasoline-powered rotary mower and a battery-powered rotary mower. Each mower worked on two different turfgrass species (bermudagrass and tall fescue) fertilized with two different nitrogen rates (100 and 200 kg ha$^{-1}$). The battery-powered mower was set at its lowest and highest blade rpm value, while the gasoline-powered mower was set at full throttle. From the data acquired, it was possible to see that the gasoline-powered mower had a much higher primary energy requirement, independent of the turf species. Moreover, comparing the electricity consumption of the battery-powered mower over time, it was possible to see that the power consumption varied according to the growth rate of both turf species. These results show that there is a partial waste of energy when using a gasoline-powered mower compared to a battery-powered mower.

**Keywords:** rotary mowing; mower comparison; energy saving

## 1. Introduction

Mowing is one of the major tasks concerning turfgrass management [1,2] and it is a very important operation, especially for sports turfs [3]. Moreover, mowing is also one of the greatest stresses that a turfgrass will endure since it removes part of the photosynthetic leaf area [2], and it should be carried out considering the specific features of each turfgrass species [1]. Turf species can be subdivided in cool-season turf species and warm-season turf species. Warm season turf species require less water to produce the same amount of dry matter when compared to cool season turf species, so they can adapt better to Mediterranean climates [4–6]. Some warm-season turf species are extremely hard to mow: zoysiagrass (*Zoysia spp.*) and bermudagrass (*Cynodon spp.*) [5]. Mowing hard-to-mow sports turf species requires a higher frequency of mower blade sharpening [7]. In fact, to have the best turf quality, mower blades need to perform a clean cut without shredding the leaves [2,8,9]. Dull mower blades can increase fuel consumption, as observed by Steinegger et al. [10], who found that gasoline consumption increased by 22% using these kind of blades. Based on their operating principle, turfgrass mowers are generally divided in rotary mowers and reel mowers. Large areas where turf quality is not a target may be mown using flail mowers. Whether or not flail mowers can equal the mowing quality of reel mowers or rotary mowers is unclear. Some authors [1] claim that flail mowers cannot equal the mowing quality of reel mowers nor rotary mowers, while Parish and Fry [11] have observed that, if a flail mower is properly sharpened, it may produce the same turf quality and mowing quality of a rotary

mower. In Italy, the maintenance of home lawns is usually carried out with rotary mowers [12]. Rotary mowers can be powered by gasoline engines or by electric engines. Electric rotary mowers are usually supplied using an electric cord or a battery [12]. Until a few years ago, the only electric rotary mowers used for private lawns in Italy were supplied with a cord. In fact, battery-powered rotary mowers are very innovative machines; however, they are still not widespread because they are more expensive than cord-supplied models, and they will cover a surface between 500 and 1000 m$^2$ [12]. Gasoline-powered rotary mowers are not designed for a precise rpm adjustment of the mowing blade, so the trend is to use them at full throttle. Instead, more innovative rotary mowers, such as battery-powered rotary mowers, can help to set the cutting blade rpm speed at a precise value. Fluck and Busey [13] carried out a trial aiming to compare a cord-supplied electric rotary mower and a gasoline-powered rotary mower, the results of which highlighted that the gasoline-powered rotary mower had a much higher primary energy (energy from primary sources transformed into electric energy) requirement than the cord-supplied electric mower. Moreover, the primary energy requirement of the electric mower showed some variation depending on turfgrass species and nitrogen fertilization rates. Unfortunately, whether or not using a battery-powered rotary mower rather than a gasoline-powered rotary mower can help saving energy is still unclear. The aim of this trial was to compare the energetic aspects of battery-powered and gasoline-powered rotary mowers working on different turfgrass species. The trial was carried out to simulate the maintenance of a high quality sports turf in order to determine the energetic aspects of the different mowing systems.

## 2. Materials and Methods

The experimental trial was carried out in S. Piero a Grado, Pisa (43°39′ N, 10°21′ E, 5 m a.s.l.) from May to November 2017 on a two-year old stand of *Festuca arundinacea* cv Grande and on a 14-month-old stand of *Cynodon transvaalensis* x *Cynodon dactylon* hybrid cv Patriot. The stand was established on a soil characterized by the following physical-chemical properties: 90% sand, 6% silt, 4% clay, pH 6.6, 1.4 g kg$^{-1}$ of organic matter; EC 0.44 dS m$^{-1}$, water availability 3.50 % *w/w*. In April, a two-way randomized blocks experimental design (A × B) with three replications was adopted. Factor (A) consisted of three different mowing systems: 1) manual mowing with a Honda mod. HRD 536 HX (Honda France Manufacturing, Ormes, France) walk-behind gasoline rotary mower with a blade revolving speed of 2800 rpm; 2) manual mowing with a Pellenc mod. Rasion Smart (Pellenc, Pertuis, France) walk-behind battery powered electric mower with a blade revolving speed of 3000 rpm; and 3) manual mowing with a Pellenc mod. Rasion Smart walk-behind battery powered electric mower with a blade revolving speed of 5000 rpm. Factor (B) consisted of two nitrogen rates (100 and 200 kg ha$^{-1}$) applied on May 9 and on August 21 with a rotary spreader using ammonium sulphate (21-0-0). Battery specifications and engine specifications of the two mowers are shown in Table 1. Working speed was 3 km h$^{-1}$. Working width was 60 cm for the battery-powered mower and 53 cm for the gasoline-powered mower. All mowers were equipped for clipping removal. The blades of all mowers were sharpened every three weeks.

For each of the two turf species, the area was 216 m$^2$ (6 × 36 m) subdivided in three randomized blocks, each of 72 m$^2$ (6 × 12 m). Plots were mowed once per week. Mowing height was 3.5 cm. Irrigation was applied as necessary.

Every three weeks, the following parameters were assessed and determined:

- gasoline consumption using a measuring cylinder; and
- electricity consumption by recording real time energy consumption indicated on the display of the battery-powered mower;

In order to record the real time energy consumption of the battery-powered mower, a camera was placed close to the display located on the side of the battery. As the battery-powered mower started working, this display continuously showed the real time electricity consumption of the machine measured in watts (w).



**Table 1.** Battery specifications and engine specifications of the battery-powered mower and gasoline-powered mower.

| Parameter | Unit | Value |
|---|---|---|
| **Battery-powered mower (li-ion battery)** | | |
| Tension | V | 43.6 |
| Battery capacity | Ah | 25.6 |
| Energy storage | Wh | 1100 |
| Max power | W | 1700 |
| Average lifespan | Recharging cycles | Over 1000 |
| Charging time | h | 10 |
| **Gasoline-powered mower (four-stroke engine)** | | |
| Number of cylinders | number | 1 |
| Engine displacement | $cm^3$ | 160 |
| Power output | kWh | 2.7 |
| Average fuel consumption | L/h | 0.8 |

## 3. Results

The primary energy requirement of the machines has been calculated considering the efficiency of the Italian National Electric System, that is 0.546 [14], the gasoline heating value, equal to 9.2 kWh/L [15], and the charging efficiency of li-ion batteries [16], equal to 91%. The total average mowing time was 360 sec for the gasoline-powered mower and 660 sec for the battery-powered mower. The average field consumption of both mowers showed some variation from tall fescue to bermudagrass (Table 2). Both mowers had a higher average field consumption when working on bermudagrass rather than on tall fescue (Table 2). The average field consumption of the gasoline-powered mower showed less variation from bermudagrass to tall fescue compared to the variation of average field consumption of the battery-powered mower.

**Table 2.** Field consumption of the gasoline-powered mower at full throttle and of the battery-powered mower at 5000 rpm on both turf species.

| Machine | Average Field Consumption | |
|---|---|---|
| | **Tall Fescue** | **Bermudagrass** |
| Battery-powered mower | 2.72 kWh/ha (electric energy) | 4.63 kWh/ha (electric energy) |
| Gasoline-powered mower | 5.24 L/ha (gasoline) | 5.59 L/h (gasoline) |

The primary energy requirement, derived from average field consumption, showed the same trend (Table 3). The battery-powered mower showed more variation in the primary energy requirement compared to the gasoline-powered mower when working on bermudagrass rather than on tall fescue (Table 3).

**Table 3.** Primary energy requirement of the gasoline-powered mower at full throttle and of the battery-powered mower at 5000 rpm on both turf species.

| Machine | Primary Energy Requirement | |
|---|---|---|
| | **Tall Fescue** | **Bermudagrass** |
| Battery-powered mower | 5.47 kWh/ha | 9.32 kWh/ha |
| Gasoline-powered mower | 48.21 kWh/ha | 51.42 kWh/ha |

Figure 1 shows the power consumption mean value of the battery-powered mower working on tall fescue during the whole trial. The power consumption of the battery-powered mower shows higher values when N rates are higher and when blade speed is higher (Figure 1).

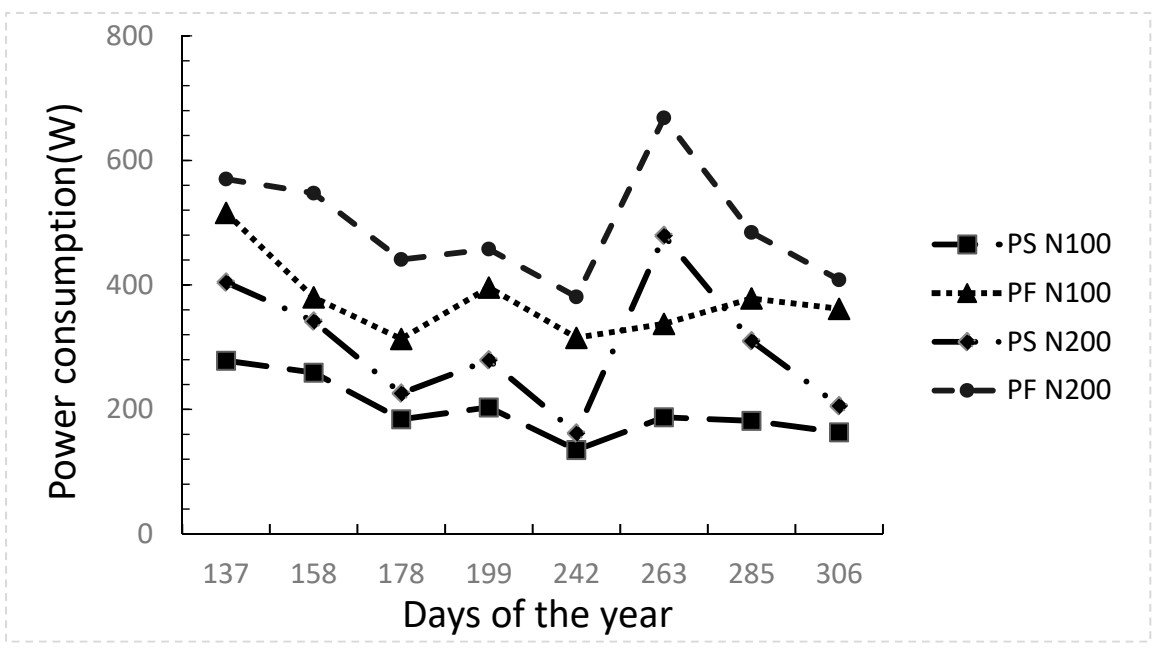

**Figure 1.** Power consumption mean value of battery-powered mower working on tall fescue on 17 May, 7 June, 27 June, 18 July, 30 August, 20 September, 12 October, and 2 November. PS = Battery-powered mower revving at 3000 rpm, PF = Battery-powered mower revving at 5000 rpm.

However, even though the power consumption mean value of the battery-powered mower shows higher values when N rates are higher and when blade speed is higher (Figure 1), the trend of the power consumption curves is similar to the trend of tall fescue growth rate throughout the year.

Figure 2 shows the power consumption mean value of the battery-powered mower working on bermudagrass during the whole trial. The power consumption of the battery-powered mower shows higher values when N rates are higher and when blade speed is higher (Figure 2).

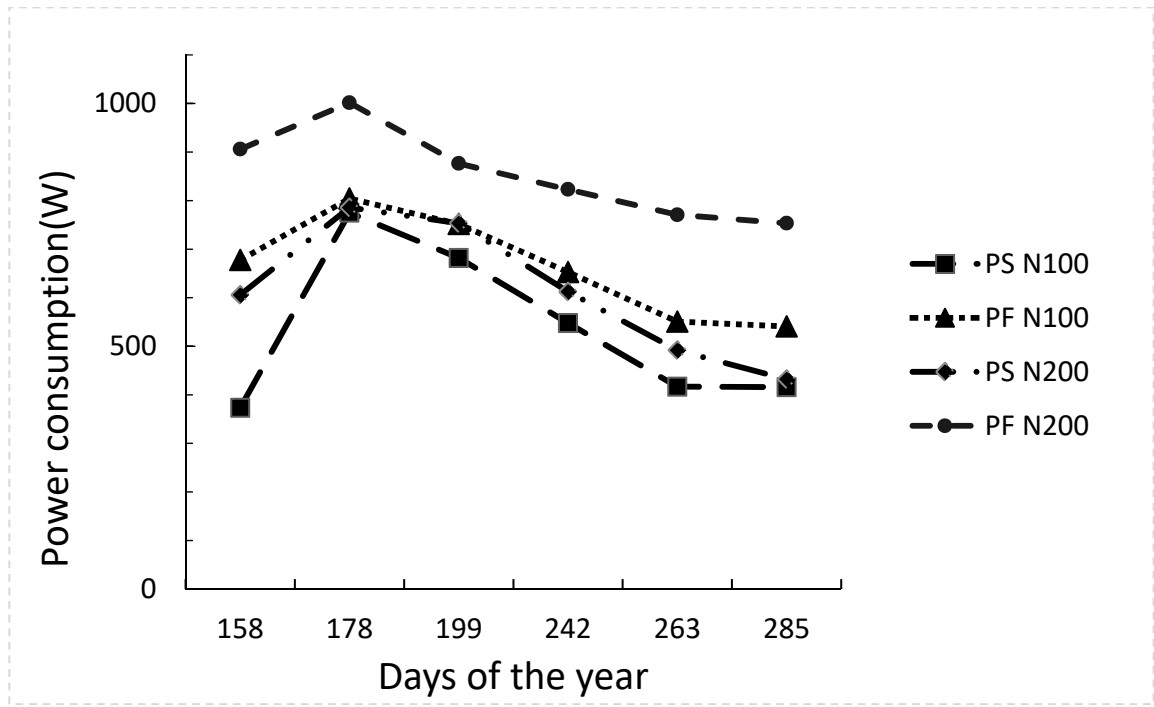

**Figure 2.** Power consumption mean value of battery-powered mower working on bermudagrass on 7 June, 27 June, 18 July, 30 August, 20 September, and 12 October.

As previously seen on tall fescue, the trend of the power consumption curves is similar to the trend of bermudagrass growth rate throughout the year, independent of blade speed or nitrogen fertilization.

## 4. Discussion

The battery-powered mower had a much lower primary energy requirement compared to the gasoline-powered mower, on both turf species. The average field consumption—and, consequently, the primary energy requirement—of the gasoline-powered mower showed a small variation when the mower worked on bermudagrass rather than on tall fescue (5.59 L/ha vs 5.24 L/ha). The battery-powered mower, instead, showed a much larger variation of average field consumption (and primary energy requirement) when the mower worked on bermudagrass rather than on tall fescue (4.63 kWh/ha vs 2.72 kWh/ha). A similar trend was observed by Fluck and Busey [13], who observed a different consumption of the electric mower working on different turf species. In the present trial, a possible reason of this difference is that, whereas the gasoline-powered mower constantly worked at full throttle during the whole trial, the power delivery of the battery-powered mower was automatically adjusted by its smart management system. In fact, the field consumption of the battery-powered mower followed the growth rate of both turfgrass species during the whole trial as shown in Figures 1 and 2. Moreover, the difference in primary energy requirement between the gasoline-powered mower and the battery-powered mower was very large (see Table 3). Dull blades may increase the power required for mowing, as observed by Steinegger et al. [10], who found that dull mower blades increase fuel consumption by 22%. However, in this trial the blades of both machines were always kept properly sharpened, so the primary energy requirement could not increase because of blade sharpness. Fluck and Busey [13] also found a large difference in the primary energy requirement between the gasoline-powered mower and the battery-powered mower. The gasoline-powered mower used by Fluck and Busey [13] required 6.32 times more energy working on *Stenotaphrum secundatum* and 7.55 times more energy working on *Paspalum notatum* than the cord-supplied electric mower. In the present trial, the higher power output (2.7 kW) and the lower efficiency (25% approximately) of the gasoline engine compared to the lower power output (1.7 kW) and the higher efficiency (95% approximately) of the electric engine of the battery-powered mower [17,18] probably contributed to enhance the difference in primary energy requirements. Using the gasoline-powered mower at full-throttle may also have further increased the difference of primary energy consumption of the two mowers (battery-powered mower 9.32 kWh/ha; gasoline-powered mower 51.42 kWh/ha.).

## 5. Conclusions

Based on what has been observed during this trial, it is possible to say that battery-powered mowers save energy compared to gasoline-powered mowers. This lower energy requirement not only depends on the higher efficiency of electric engines compared to gasoline engines. In fact, since an optimal mowing quality is the aim of turfgrass mowing, high mower blade rpm is important. For this reason, gasoline-powered mowers need to be used at full throttle to keep the blade revving at the highest possible speed. Instead, battery-powered mowers can adjust the energy consumption depending on mowing effort, independent of the mower blade rpm. As a consequence, using a gasoline-powered rotary mower will result in a partial waste of power if the aim is to have a high quality turf. Moreover, battery-powered mowers are more versatile, help to save local pollution, and reduce noise emissions when compared to gasoline-powered mowers.

**Author Contributions:** M.P., M.F., F.L., C.F., L.M., M.R., A.P., M.G., S.M., L.C., M.V., N.G., M.S. conceived and designed the experiments, performed the experiments, analysed the data, contributed analysis tools and wrote the paper.

**Funding:** This research received no external funding.

**Acknowledgments:** We would sincerely like to thank Pellenc Italia S.r.l. (Colle di Val d'Elsa, Siena, Italy) for providing the battery-powered mower, technical assistance and supporting this research.

**Conflicts of Interest:** The authors declare no conflict of interest.

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
