# Peer review of "Energetic Aspects of Turfgrass Mowing: Comparison of Different Rotary Mowing Systems"

_agriculture, doi:10.3390/agriculture9080178_

Round 1

Reviewer 1 Report

In this work, authors determined and compared the power consumption of three different rotary mowers consisting of battery-powered and gasoline-power mowers. I have a few comments as following:

1) Please present the battery specifications and engine specifications for the battery-powered mower and gasoline-power mower, respectively, in a table in the Materials and Methods section.

2) Please define primary energy requirement.

3) In the Results section, you may present the average time of mowing operation lasted for different mowers in this study.

4) What deficiencies did you find out for this work that needs further research?

5) The Conclusion section needs to be re-written. Your current conclusion is basically part of Discussion section. You may present the main technical findings of your research in the conclusion section.

Author Response

Dear Reviewer,

we would like to kindly thank you for your precious work and for all the useful suggestions you gave us to help improve this manuscript. We did our best to carry out all the changes to this paper according to your suggestions.

Best regards.

1) Please present the battery specifications and engine specifications for the battery-powered mower and gasoline-power mower, respectively, in a table in the Materials and Methods section.

Done. All the specifications have been presented in Table in the M&M section.

2) Please define primary energy requirement.

Done. It has been implemented in the text in the Introduction section “(energy from primary sources transformed into electric energy)”.

3) In the Results section, you may present the average time of mowing operation lasted for different mowers in this study.

Done. Mowing time has been presented in the Results section.

4) What deficiencies did you find out for this work that needs further research?

We did not find big deficiencies requiring further research. This kind of study may be repeated if major changes will be performed to gasoline-powered mowers trying to improve their efficiency.

5) The Conclusion section needs to be re-written. Your current conclusion is basically part of Discussion section. You may present the main technical findings of your research in the conclusion section.

Done. Conclusions have been rewritten looking trying to explain the main findings.

Reviewer 2 Report

This is an interesting and timely contribution. However, some shortcomings I have found. The theoretical foundations of mowing machines work were elaborated in great detail by Sverker Persson in an ASAE Monograph “Mechanics of Cutting Plant Materials” (American Society of Agricultural Engineers, 1987, St. Joseph, Michigan, USA, ISBN 0-916150-86-0). All theoretical considerations related to cutting, including energy demands and the effects influencing it are discussed in this book. RPM and blade sharpness are essential. I lack this theoretical background in the paper. In my opinion, it would certainly contribute to a better discussion and understanding of presented research.

On p. 3, it is stated that battery-powered mower had a much lower primary energy requirement compared to the gasoline-powered mower. This fact would be better explained in Discussion section. It is also clear from Figure 1 and Figure 2 that battery-powered mower showed lower power demands when working with smaller RPM.

Last comment I have on the formal processing of Figure 1 and 2. It is not possible to connect the measured values with rounded curves as in these figures. The course of measured values between individual measured points is not known, nor in its surroundings. I highly recommended to connect the points only with a polyline, straight between individual points.

Author Response

Dear Reviewer,

we would like to kindly thank you for your precious work and for all the useful suggestions you gave us to help improve this manuscript. We did our best to carry out all the changes to this paper according to your suggestions.

Best regards.

1) The theoretical foundations of mowing machines work were elaborated in great detail by Sverker Persson in an ASAE Monograph “Mechanics of Cutting Plant Materials” (American Society of Agricultural Engineers, 1987, St. Joseph, Michigan, USA, ISBN 0-916150-86-0). All theoretical considerations related to cutting, including energy demands and the effects influencing it are discussed in this book. RPM and blade sharpness are essential. I lack this theoretical background in the paper. In my opinion, it would certainly contribute to a better discussion and understanding of presented research.

Done. Thank you for this great suggestion, we cited this book in the Introduction section.

2) On p. 3, it is stated that battery-powered mower had a much lower primary energy requirement compared to the gasoline-powered mower. This fact would be better explained in Discussion section. It is also clear from Figure 1 and Figure 2 that battery-powered mower showed lower power demands when working with smaller RPM.

Done. The phrase has been moved to the Discussion section.

3) Last comment I have on the formal processing of Figure 1 and 2. It is not possible to connect the measured values with rounded curves as in these figures. The course of measured values between individual measured points is not known, nor in its surroundings. I highly recommended to connect the points only with a polyline, straight between individual points

Done. Thank you for having shown us this important error! The figures have been created again with points connected by straight lines.

Round 2

Reviewer 2 Report

Article was improved.

Author Response

Dear Reviewer 2,

thank you for the contributions you gave us to improve this article, we hope that with the last changes performed it will be even more valuable.

Best regards

Michel Pirchio